# Comparison of Anti-Bacterial Efficacy between Passive Ultrasonic Irrigation and 980 nm-GaAlAs Laser Application in Two Root Types

**DOI:** 10.3390/medicina57060537

**Published:** 2021-05-27

**Authors:** Sung-Eun Yang, Yong-Min Kim

**Affiliations:** Department of Conservative Dentistry, Seoul St. Mary’s Hospital, College of Medicine, The Catholic University of Korea, Seoul 06591, Korea; yongmin32@nate.com

**Keywords:** antibacterial efficacy, passive ultrasonic irrigation, GaAlAs laser, *Enterococcus faecalis*

## Abstract

*Background and Objectives:* The purpose of the present study was to investigate the anti-bacterial efficacy of passive ultrasonic irrigation (PUI) and a 980-nm-gallium-aluminum-arsenide (GaAlAs) laser using a real-time DNA-based quantitative polymerase chain reaction (qPCR) assay and scanning electron microscopy (SEM). *Materials and Methods*: Eighty-six extracted single- and double-rooted human teeth were used in the experiment. The following four experimental groups were evaluated, as well as a control group: group 1: single root + PUI (*n* = 20); group 2: single root + laser application (*n* = 20); group 3: double roots + PUI (*n* = 20); group 4: double roots + laser application (*n* =20); control group (*n* = 6): 3 single roots, 3 double roots. The qPCR assay was performed in three stages to evaluate the efficacy of the adjunctive method against *Enterococcus faecalis*. SEM analysis was also used to examine the microstructure of root canal surfaces. The data were statistically analyzed using the Friedman test and the Kruskal–Wallis test with Bonferroni adjustment. *Results*: The decrease in the DNA levels from S1 (before preparation) to S2 (just after preparation) was highly significant in all groups, and decreases in DNA levels from S2 to S3 (after incubation for 1 week) were found in all experimental groups except group 1. An inter-group analysis showed that PUI was significantly more effective in terms of antibacterial efficacy than GaAlAs laser irradiation in single root (*p* < 0.05). However, in double roots, experimental groups did not show significantly lower DNA levels than the control group (*p* > 0.05). SEM images showed that cleaning of the root canal surface and reduction of dentin debris were achieved only in single-rooted teeth when using PUI application. *Conclusions*: Two adjunctive methods were effective in reducing *E. faecalis* in single rooted tooth.

## 1. Introduction

The root canal system has various complex structures, such as isthmuses, ramifications, deltas, and accessory canals, which can result in incomplete debridement of bacteria and their by-products when infected root canals are cleaned using traditional irrigation delivery systems [1]. A number of irrigation methods and protocols have been introduced to accomplish the major goal of root canal therapy, which is the complete elimination of the irritant and debris generated during biomechanical instrumentation. However, many authors have reported that this goal is very difficult to achieve [2,3,4,5].

Passive ultrasonic irrigation (PUI) has been widely used since its introduction by Weller as a way to remove irritants more effectively [6]. Previous studies have demonstrated that ultrasonic irrigation showed better antibacterial efficacy than traditional chemomechanical irrigation using only a plastic syringe and needle [3,7]. Ultrasonic waves are transmitted from the ultrasonic device through the tip, resulting in an acoustic streaming and cavitation effect that enables more thorough cleaning and disinfection of the root canal system. Grundling et al. demonstrated that PUI can be an aid in root canal cleaning [8].

Recently, gallium-aluminum-arsenide (GaAlAs) semiconductor lasers have been introduced for endodontic procedures [9,10,11,12]. The advantages of GaAlAs semiconductor lasers include their non-abrasive nature, meaning that they are expected to minimize damage to the dentinal tubule, and their ability to penetrate through the dentinal tubule more deeply, up to 1000 μm. Few studies have investigated the antibacterial efficacy of this system, and the results remain inconclusive [13,14].

The purpose of this in vitro study was to investigate the antibacterial efficacy of PUI and a GaAlAs laser irradiation on an artificially inoculated teeth model (in single-rooted and double-rooted teeth) using a real-time DNA-based quantitative polymerase chain reaction (qPCR) assay. Additionally, scanning electron microscopy (SEM) was used to examine the morphological features of the surfaces of root canals after application of two different adjunctive methods.

## 2. Materials and Methods

### 2.1. Sample Preparation

This study was approved by the Institutional Review Board of Seoul St. Mary’s Hospital, the Catholic University of Korea (KC18TNSI0037). Eighty-six extracted human teeth were selected and stored in vials containing sterile saline solution. Lower premolars with single root canals and upper premolars with double root canals for upper ones were selected for this study. Before the teeth were used in the experiment, they were soaked in 2.5% sodium hypochlorite (NaOCl) solution to remove residue from the teeth for a period not exceeding 24 h. The samples were then sterilized at a temperature of 121 °C with a pressure of 118 kPa in an autoclave for a period of 30 min.

Next, the working length was set to 1 mm shorter than when the file was passed through the apical foramen using a #10 K-file (Dentsply Maillefer, Ballaigues, Switzerland). To unify the apical size, all roots were cleaned and shaped using Protaper instruments (Dentsply Maillefer) up to F1, which was applied at the working length. After the mechanical preparation was completed, 1 min of irrigation was performed, using 2.5% NaOCl and 17% ethylenediaminetetraacetic acid (EDTA) for each canal with a 28-G side-vented needle (Monoject endodontic needle; Tokyo/Kendall, Mansfield, MA, USA). The flow rate was approximately 0.1 mL/s. Sealing of the apical foramen to prevent bacterial leakage was performed with a manicure and light-cured composite resin (Z250, 3M ESPE, Seefeld, Germany).

In order to facilitate easier handling and identification of samples, a 12-well cell culture microplate was used. The teeth were set up vertically using a silicone impression material, and samples were then randomly assigned to one of five groups (described later).

### 2.2. Bacterial Preparation and Inoculation

*Enterococcus faecalis* (Schleifer and Kilpper-Balz ATCC^®^ 29212™) was grown on brain heart infusion (BHI) agar plates. The bacterial cells were suspended in 10 mL of BHI broth and incubated at 37 °C for 18 to 24 h in a bacteriological incubator.

In each of the 86 samples, which had been previously sterilized, 20 μL of the inoculum was injected into each canal using a micropipette under aseptic conditions of a clean bench. Next, manual agitation was performed at the working length for 15 s using a 15 K-file so that the inoculum could be transferred to the apical part of the root more effectively. Subsequently, the 12-well plates were sealed with para-film and incubated at 37 °C in 100% humidity for 14 days. Fresh bacterial medium (20 μL) was inoculated into each canal every 2 days.

### 2.3. Classification of the Groups

The treatment protocol is presented as a flowchart in Figure 1. The groups were distributed as follows:Single root + PUI group (*n* = 20): PUI was performed using a piezoelectric ultrasonic device at a power setting of 2 (Satelec, Acteon, Merignac, France) with an Irrisafe tip (15/K size, Acteon, Mt. Laurel, NJ, USA) in an in-and out motion for up to two-thirds of the length of the canal for a total of 1 min.Single root + laser application group (*n* = 20): Before irradiation with a GaAlAs diode laser (Hulaser, Seoul, Korea), the canal was fully dried with paper points (Dentsply Maillefer). Then, the canal was irradiated with a pulsed 980-nm GaAlAs diode laser at an output power of 1 W at 50 Hz. During the laser irradiation, a slow helical movement method was applied in the canal with a 200-μmdiameter fiber tip for 10 s and a 10-s break for up to two-thirds of the length of the canal for a total of 1 min.Double roots + PUI group (*n* = 20): The protocol was similar to group 1, except that each tooth in this group contained two root canals.Double roots + laser application group (*n* = 20): The protocol was similar to group 2, except that each tooth in this group contained two root canals.Control group (*n* = 3 single root, 3 double roots): The root canals were irrigated with sterile saline solution, not with 2.5% NaOCl.

### 2.4. Microbiological Analysis

Before the respective adjunctive method was applied, S1 (before preparation) sampling was performed using three #20 sterile paper points (Dentsply Maillefer) for each canal, which were stirred with a 15 K-file (Dentsply Maillefer) for 10 s after soaking up the canal contents for 1 min. The soaked paper points were transferred to S1 sampling vials containing 200 μL of phosphate buffered saline (PBS) for quantitative polymerase chain reaction (qPCR) analysis.

Final shaping and chemomechanical debridement of the canal were performed using sterile ProFile Ni-Ti files (Dentsply Maillefer) in a rotary crown-down technique up to #30/0.06 for double roots, and #40/0.06 for a single root. Between the mechanical preparations, 1 min of irrigation with 2.5% NaOCl and 17% EDTA was done, followed by a final 1 min flush with 2.5% NaOCl through a 28G side-vented needle to 2 mm short of the working length. The flow rate was approximately 0.1 mL/s. Next, the adjunctive method was applied to each group, as described above. Before the S2 (just after preparation) specimens were obtained, the root canal was flushed with 5% sodium thiosulfate for 30 s in order to neutralize the NaOCl. S2 sampling was done in the same way as the S1 sample described above. The control group (*n* = 6) was irrigated with a 3 min rinse of sterile saline solution only, with a flow rate of 0.1 mL/s, followed by a 30-s rinse of 5% sodium thiosulfate, and the S2 sampling was done. The total irrigant contact time was 3.5 min.

To simulate clinical conditions, samples were flushed with 2.5% NaOCl for 1 min (except the control group, which instead received sterile saline irrigation for 1 min), sealed with Caviton (GC, Tokyo, Japan), and incubated for one week at 37 °C, 95% humidity. One week later, S3 (after incubation for 1 week) sampling was performed in the same way as the S2 sampling. After the series of S1, S2, and S3 sampling, two teeth from each group were immediately immersed in 4% paraformaldehyde for fixation and subsequent analysis using scanning electron microscopy (Hitachi High-Technologies, Tokyo, Japan).

### 2.5. DNA Extraction and qPCR Analysis

Genomic DNA was extracted using the AccuPrep Genomic DNA Extraction Kit (Bioneer Co., Daejeon, Korea) according to the manufacturer’s instructions. The presence of bacteria was verified in the sample by using a 16S rDNA species-specific primer pair. The qPCR was performed using MyiQ Real-Time PCR Detection System (Bio-Rad, Hercules, CA, USA). For all experiments, 1 mL of extracted genomic DNA was added to 19mL of PCR mixture containing SYBR Green (iTaq™UniversalSYBR^®^ Green Supermix; Bio-Rad), 10 pmol of each primer (Bioneer Co.), and thrice-distilled water. The cycling parameters for quantification were as follows: 95 °C for 5 min, 40 cycles of denaturation at 95 °C for 5 s, and annealing at 55 °C for 30 s. For the melt-curve analysis, the temperature was increased in increments of 0.2 °C from 65 to 95 °C. Melting peaks were used to determine the specificity of the qPCR. The efficacy of the adjunctive method against *E. faecalis* was calculated based on the cycle threshold (C(t)) value. The DNA level was then calculated using normalization of the C(t) value to investigate relative quantification analysis as described by Livak et al. [15].

### 2.6. Preparation for Scanning Electron Microscopy

Samples were fixed for 2 days in 4% paraformaldehyde and then washed three times for 30 min each using a combined solution of 0.2 mol/L phosphate buffer and distilled water at a ratio of 1:1. Then, the samples were dehydrated by immersion in acetone (in staged 30%, 50%, 70%, 90%, and 100% solutions). Longitudinal sections of samples were taken using a diamond saw, taking care not to invade the inner part of the root canal. The samples were then coated with gold-palladium to promote conductivity.

The evaluation was performed using SEM at a magnification of ×2000, dividing the root into thirds.

### 2.7. Statistical Analysis

The average and standard deviation of the DNA levels of *E. faecalis* were calculated using SAS for Windows version 9.4 (SAS Corp, Cary, NC, USA). The Friedman test was used for intragroup analysis (S1, S2, and S3), and the Kruskal–Wallis test with Bonferroni adjustment was used for intergroup analysis of the S1, S2 and S3 data. The level of significance was set at *p* = 0.05.

## 3. Results

### 3.1. Microbiological Analysis

The analysis of S1 samples revealed no significant difference between each group, indicating that a homogeneous and reliable baseline was achieved. Consequently, data from S2 and S3 could be used for direct intra- and inter-group comparisons. Figure 2 shows the C(t) value obtained for all groups and Table 1 presents the mean and standard deviation of DNA levels observed for single- and double-rooted teeth.

The decrease in the DNA levels from S1 to S2 was highly significant in all groups, and decreases in the DNA levels from S2 to S3 were found in all experimental groups except group 1. An inter-group analysis showed that PUI was significantly more effective in terms of antibacterial efficacy than GaAlAs laser irradiation in single roots (*p* < 0.05). However, in double roots, experimental groups did not show significantly lower DNA levels than the control group (*p* > 0.05).

Fold changes between the experimental and control groups are shown in Figure 3. Groups 1 and 2, which were composed of single-rooted teeth, revealed a significantly lower DNA levels than did the groups with double-rooted teeth in S2.

### 3.2. SEM Analysis

Figure 4 illustrates the SEM analysis of the surface of root canal walls after the application of each adjunctive method. In single-rooted teeth, PUI had a significant effect on cleaning of the root canal surface and reduction of dentin debris, but not in the apical third of the canal. In double-rooted teeth where the isthmus region connecting the two canals was observed, PUI was not effective for surface cleaning or debris removal in any areas, including the apical third area. 

Laser irradiation had no effect on cleaning the root canal surface and reducing dentin debris in both single-rooted teeth and isthmus area of double-rooted teeth. Interestingly, root surface changes including surface melting were not significantly noticeable in most laser groups.

## 4. Discussion

The present study conducted a comparative evaluation of the antibacterial efficacy of two adjunctive methods (PUI vs. GaAlAs laser) on single- and double-rooted teeth using a qPCR assay and SEM analysis, whereas the majority of previous studies analyzed only in single-rooted teeth [16]. The current study used qPCR assay with the C(t) value parameter, which refers to the cycle at which fluorescence achieves a defined threshold; therefore, this is a sensitive parameter for *E. faecalis* detection and quantitation of bacterial levels. An analysis of data from real time qPCR experiments may be conducted using either relative or absolute quantification. Relative quantification analysis may be easier to perform than absolute method because the use of standard curves is not required. The results of the current study showed that the two different adjunctive methods were more effective for reducing bacterial populations than the control group with a high volume of saline as the irrigant, though the effect is limited in complex structures such as isthmus area. This finding is in accordance with the results of Afkhanmi et al. and Gutknecht et al., who demonstrated the efficacy of high-power lasers for reducing *E. faecalis* and aligns with those of a previous study reporting that PUI was efficacious [8,14,17]. However, the control group in the current study, which was irrigated with sterile saline, also showed a certain degree of effectiveness in eliminating bacteria. In this study, the canals were enlarged to a profile size of a #40/0.06 taper in single-rooted teeth and a #30/0.06 taper in double-rooted teeth. The mechanical canal preparation and the process of irrigating the root canal with saline solution seemed to contribute to the antibacterial effect to some extent.

Card et al. evaluated the effectiveness of increased apical enlargement for reducing intracanal bacterial load and found that increasing the apical preparation size could lead to eradication of microorganisms more effectively in infected canals [18]. However, the apical preparation size can only be increased to a certain extent because the original root canal anatomy and canal preparation size were not variables in the present study.

Several studies have demonstrated antibacterial efficacy using various laser systems [12,19,20]. However, the effectiveness of these laser systems has been evaluated only in single, straight canals, although many teeth requiring root canal treatment in clinical settings have more varied root anatomy and complex root canal systems for which needle irrigation alone may have a limited cleaning effect [1]. In order to reflect this, the sample of the present study included maxillary premolars with two canals, including the isthmus. 

Our findings showed that both adjunctive methods seemed to be effective for reducing *E. faecalis* in the coronal and middle third areas. However, clogging caused by debris remained after PUI and laser irradiation in the apical third of the canal and isthmus area. This might have been because PUI created intense and circular fluid movement around the part of the instrument that was closer to the tip area. In the present study, the tip was applied for up to two-thirds of the length of the canal to avoid extrusion, according to manufacturer’s instruction. Therefore, it was not sufficient to clean the apical third of the root canal. In contrast, the coronal and middle portions of the root canal provided a sufficient space for exchange of the irrigant without any obstacles. These findings are in agreement with those reported by Donnermeyer et al. [21]. Similarly, GaAlAs laser application may not have been effective in killing microorganisms because the helical motion, which resulted in a spiral pattern of irradiation when used to irradiate infected root canals, did not touch the apical third region.

The results of the S3 specimens corresponded to changes in the bacterial count in multi-visit root canal treatment, not single-visit root canal treatment. Strictly speaking, the current in vitro study did not completely replicate the clinical environment. However, it was noteworthy that *E. faecalis* count decreased from S2 to S3, except for group 1. It is reasonable to propose that if orifice sealing has been done properly, in multi-visit root canal treatment, bacterial re-contamination is unlikely to occur if proper chemomechanical preparation has been done.

Recently, interest in various laser systems has increased because of the development of specialized delivery systems, including thin and flexible tips. Laser systems were previously difficult to apply in clinical conditions, and the laser beam provided irradiation only in a straight line, resulting in poor treatment efficiency in the endodontic area. The laser tip used in this study was thin (diameter: 200 μM), which enabled access to the narrow root canal space, and had a shape that could be side-fired to some extent, providing better access and the ability to penetrate further into the apical portion of the tooth.

Ribeiro et al. reported a melting and fusing appearance of root canal walls after the application of a diode laser (810 nm) at the apical region [22]. Therefore, in this study, the experiment was carried out with a relatively low strength. Thus, no melting and fusing appearance was observed on most surfaces in the SEM images. However, even after laser application, debris was not effectively removed. In many studies, laser application showed a better ability to clean debris than PUI. It may be possible that laser irradiation could exert a strong cavitation effect, leading to the formation of vapors that expand and implode. However, the apical preparation size ranged from #25 to #50, 20–150 mJ for laser output power, 200–600 μM for the diameter of the laser tip, and 2–16 mL for the total irrigant volume. These wide ranges may explain the discrepancy between the results of previous studies and the findings of the present study [23,24,25]. Furthermore, the laser used in this study was applied to the dried root canal according to the manufacturer’s instructions. This method was effective only in contact area where the laser tip was applied. However, newly introduced methods (e.g., antimicrobial photodynamic therapy and photon induced photoacoustic streaming using dye or irrigation medium would have been expected to yield a better effect on smear layer removal [26,27]. More studies are needed to compare various laser application systems.

Direct interactions between the laser and root dentin are an important consideration. The laser causes the formation of vapor bubbles, acoustic streaming, and cavitation [12]. This mechanism of laser activation can produce thermal damage to periapical tissue. Hence, the 10-s breaks between irradiation cycles in this study are needed to avoid thermal damage. No definitive conclusion has yet been reached regarding the precise maximum strength that provides a sufficient antibacterial effect in teeth, including the apical third of the canal and C-shaped canals with a long isthmus, while avoiding damage to the tooth structure or adjacent connective tissue. Future research is required to explore the potential of other adjunctive methods such as NFX (Ultradent Products, South Jordan, UT, USA) and XPF (FKG Dentaire SA, La Chaux-de-Fonds, Switzerland), and the use of intracanal medicaments, alternative disinfectants, and different concentrations of NaOCl to overcome the limitations of chemomechanical preparation for teeth with complex root canal structure [28].

## 5. Conclusions

Two adjunctive methods were effective in reducing *E. faecalis* in single-rooted teeth, but in the apical third and isthmus areas their efficacy was poor. Therefore, a combination of PUI and laser irradiation or other adjunctive methods may be considered to enhance the antibacterial effect on the dentinal tubules.

## Figures and Tables

**Figure 1 medicina-57-00537-f001:**
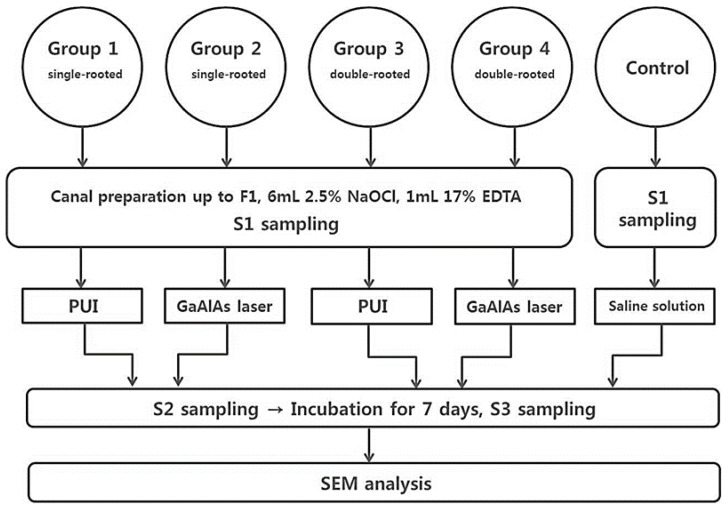
Workflow of experimental design. S1: before preparation; S2: just after preparation; S3: after incubation of 1 week; PUI: passive ultrasonic irrigation; GaAlAs: gallium-aluminum-arsenide; SEM: scanning electron microscopy.

**Figure 2 medicina-57-00537-f002:**
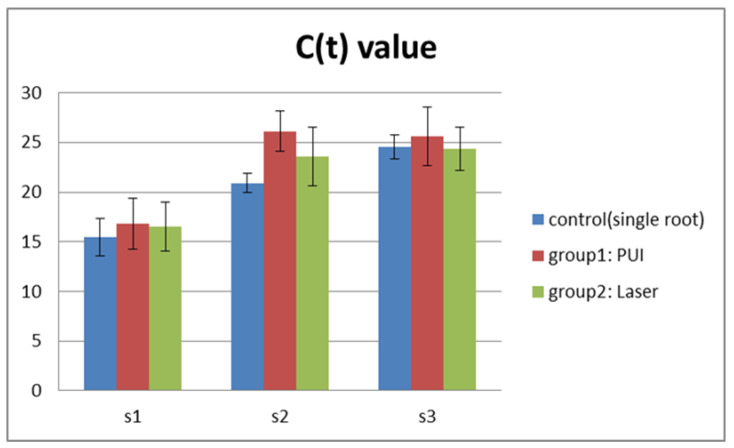
C(t) values in the experimental groups.

**Figure 3 medicina-57-00537-f003:**
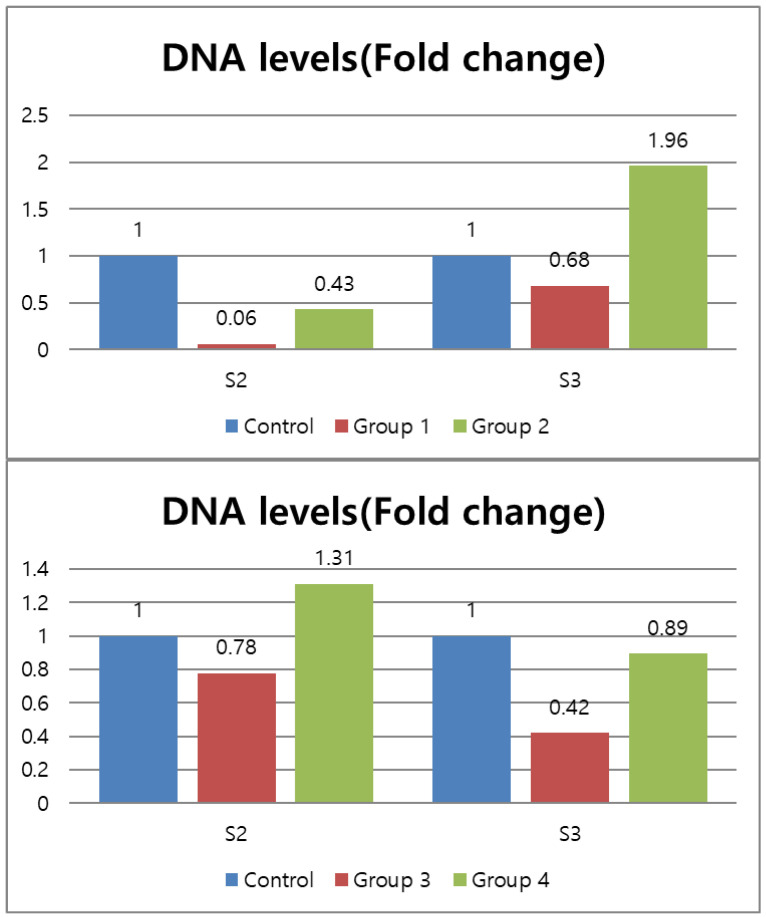
DNA levels (Fold change) of the experimental groups, calculated by the 2^−ΔΔC(t)^ method. The 2^−ΔΔCT^ method means ΔΔCT = ΔCT (experimental group) − ΔCT (control group).

**Figure 4 medicina-57-00537-f004:**
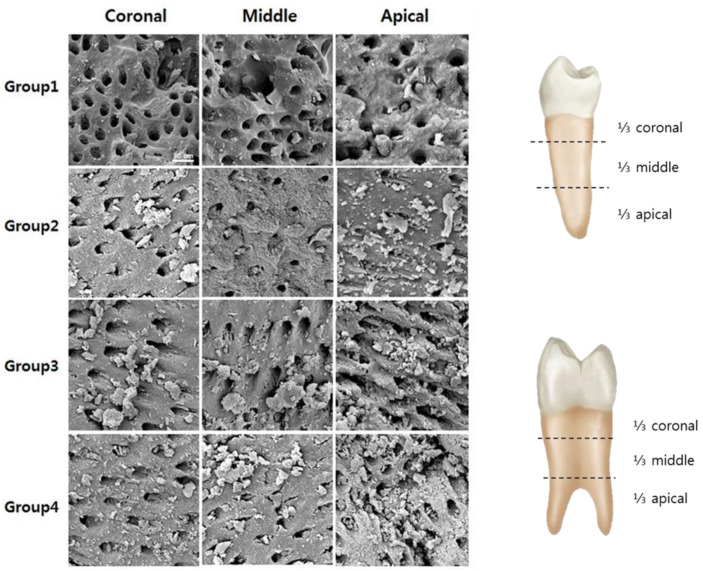
Representative SEM images from the coronal, middle, and apical thirds of the root surface showing dentinal tubules in groups 1 and 2 and isthmus area in groups 3 and 4 samples treated by different irrigation methods (×2000).

**Table 1 medicina-57-00537-t001:** Mean DNA levels (calculated using normalization of C(t) value) of the S1, S2, and S3 samples between groups (single root or double roots).

Subgroup	S1	S2	S3	*p*-Value
Group1 (*n* = 20)	0.8927 ± 3.30 ^z^	0.001 ± 0.00 ^cy^	0.001 ± 0.02 ^y^	<0.001
Group2 (*n* = 20)	0.8812 ± 1.95 ^z^	0.006 ± 0.01 ^by^	0.002 ± 0.00 ^y^	<0.001
Control (*n* = 3, single rooted)	1.0 ± 1.72 ^z^	0.013 ± 0.01 ^ay^	0.001 ± 0.00 ^x^	<0.001
*p*-value	0.99	<0.001	0.07	-
Group3 (*n* = 20)	1.0770 ± 4.52 ^z^	0.003 ± 0.01 ^y^	0.001 ± 0.00 ^bx^	<0.001
Group4 (*n* = 20)	0.8985 ± 5.10 ^z^	0.006 ± 0.10 ^y^	0.002 ± 0.00 ^ax^	<0.001
Control (*n* = 3, double rooted)	1 ± 1.91 ^z^	0.005 ± 0.00 ^y^	0.002 ± 0.00 ^aby^	<0.001
*p*-value	0.98	0.504	<0.05	-

Different superscript letters (column: a~c, row: x~z) indicate statistically significant differences between groups (*p*-value [Multiple comparison] < 0.05); S1: before preparation; S2: just after preparation; S3: after incubation of 1 week.

## Data Availability

The datasets used and/or analyzed during the current study are available from the corresponding author on reasonable request.

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
