# Peer review of "Comparison of Anti-Bacterial Efficacy between Passive Ultrasonic Irrigation and 980 nm-GaAlAs Laser Application in Two Root Types"

_medicina, 2021, doi:10.3390/medicina57060537_

Round 1
Reviewer 1 Report
Editing of English language is necessary. Some sentence are difficult to understand and should be corrected.
Please provide more details on how the canals were mechanically instrumented. The sequence of rinsing. But mostly what was the taper used. And if the final size of file was F1/20 than do You consider that this had no impact on the result? If it was indeed size 0.20 that the conclusion should be more careful.
Overall it is interesting to compare which method was more efficient. Figure 3 is not easy to draw any conclusion. Please make it easier to read.
Author Response
Thank you for taking your time to give us very helpful advices on this article.
The article has been revised according to your opinion, and please refer to the attached file to confirm the revised content.

Reviewer 2 Report
It would be interesting to compare the effects of the laser you used in your research with Nd:Yag and erbium lasers. You have stated that laser irradiation had no effect of cleaning of the root canal surface and reduction of dentin debris. Do you believe it might be due to the fact of using relatively low laser power?
Also, the glazing effects are observed usually when the smear layer has not been properly removed.
Author Response

(The authors gave the same response as above.)
